# Regulation by Different Types of Chaperones of Amyloid Transformation of Proteins Involved in the Development of Neurodegenerative Diseases

**DOI:** 10.3390/ijms23052747

**Published:** 2022-03-02

**Authors:** Vladimir I. Muronetz, Sofia S. Kudryavtseva, Evgeniia V. Leisi, Lidia P. Kurochkina, Kseniya V. Barinova, Elena V. Schmalhausen

**Affiliations:** 1Belozersky Institute of Physico Chemical Biology, Lomonosov Moscow State University, 119991 Moscow, Russia; lpk56@mail.ru (L.P.K.); kmerkushina@gmail.com (K.V.B.); shmal@belozersky.msu.ru (E.V.S.); 2Faculty of Bioengineering and Bioinformatics, Lomonosov Moscow State University, 119991 Moscow, Russia; sofiia.kudriavtceva@gmail.com; 3Faculty of Biology, Lomonosov Moscow State University, 119991 Moscow, Russia; shelarisu@gmail.com

**Keywords:** alpha-synuclein, amyloid proteins, chaperones, chaperonins, microbiota, misfolded proteins, nitrosylation, polymers, post-translational modifications, prion protein

## Abstract

The review highlights various aspects of the influence of chaperones on amyloid proteins associated with the development of neurodegenerative diseases and includes studies conducted in our laboratory. Different sections of the article are devoted to the role of chaperones in the pathological transformation of alpha-synuclein and the prion protein. Information about the interaction of the chaperonins GroE and TRiC as well as polymer-based artificial chaperones with amyloidogenic proteins is summarized. Particular attention is paid to the effect of blocking chaperones by misfolded and amyloidogenic proteins. It was noted that the accumulation of functionally inactive chaperones blocked by misfolded proteins might cause the formation of amyloid aggregates and prevent the disassembly of fibrillar structures. Moreover, the blocking of chaperones by various forms of amyloid proteins might lead to pathological changes in the vital activity of cells due to the impaired folding of newly synthesized proteins and their subsequent processing. The final section of the article discusses both the little data on the role of gut microbiota in the propagation of synucleinopathies and prion diseases and the possible involvement of the bacterial chaperone GroE in these processes.

## 1. Introduction

Some neurodegenerative diseases are usually referred to as conformational diseases, since their occurrence and development depend on the pathological transformation of structurally altered proteins [1,2]. The most striking example is prion diseases, which are characterized by the formation of amyloid structures caused by the interaction of a native prion protein with an infectious form of the prion with an incorrect conformation [3,4]. Thus, the development of neurodegenerative diseases of an amyloid nature is based on two processes: a change in the structure of an amyloidogenic protein and the formation of various aggregates from such a protein with a disturbed conformation. It is quite obvious that both processes should be affected by various chaperones that are present in the cells, since they are responsible for the correct protein folding, prevent protein aggregation, and can even destroy already formed aggregates [5]. The first hypotheses about the role of chaperones in the development of neurodegenerative diseases were quite simple. They were based on data on the ability of chaperones to prevent protein aggregation. It was assumed that an increase in the concentration of chaperones that destroy amyloid aggregates should prevent the development of neurodegenerative diseases. Therefore, one of the strategies for the treatment of amyloid diseases could be the use of externally added chaperones. Such approaches were developed up to the beginning of clinical trials for the treatment of Huntington’s disease [6]. However, in parallel with the development of such approaches, information on a more complex and controversial role of chaperones in the development of neurodegenerative diseases has been accumulating. On the one hand, it was shown that the toxicity of amyloid aggregates depends on their structure. As a rule, large fibrillar structures are not toxic, and relatively small oligomers have maximum toxicity. Therefore, the use of chaperones for the destruction of amyloid fibrils can lead to undesirable consequences due to the appearance of neurotoxic oligomeric forms of proteins. On the other hand, there is evidence that it is the interaction of some chaperones with amyloidogenic proteins that cause their pathological transformation and initiate the formation of amyloid structures [7].

Probably, such processes can be characteristic of chaperones with partially impaired functions. Thus, one should not hope that all chaperones will have a beneficial effect on the treatment of neurodegenerative diseases. The mechanisms of interaction of chaperones with various amyloidogenic proteins should be carefully studied, taking into account both the functioning of chaperones and the properties of these proteins, paying special attention to post-translational modifications of the proteins that can radically change their pathological transformation.

## 2. Types of Chaperones That May Be Associated with Pathological Transformation of Proteins

Chaperones play a key role in a wide range of processes occurring in the cells of all living organisms, from bacteriophages and viruses to plants and animals. They are involved in de novo protein folding, the assembly of oligomeric proteins, the breakdown of protein aggregates and the refolding of stress-denatured proteins, protein translocation, and the removal of aberrant proteins by their degradation with the ubiquitin–proteasome system or by autophagy [8,9,10]. Disorders in the structure and functional activity of chaperones due to mutations or post-translational modifications, changes in the content of individual chaperones or in the ratio of different types of chaperones, as well as their localization can lead to serious consequences not only for the cell but also for the whole organism. Any of these factors can cause the development of various pathologies: the so-called chaperonopathies, some forms of cancer, as well as autoimmune and neurodegenerative diseases [1,7]. The development of the latter is facilitated by age-related dysregulation of cellular proteostasis in favor of the accumulation of misfolded and aggregated proteins in cells [11].

There are several different classes of chaperones in cells, which are also called heat shock proteins (HSPs), because their production is stimulated by heat stress [12,13]. Historically, chaperones are usually classified according to the molecular mass of their polypeptide chains: Hsp40, Hsp60 (chaperonins), Hsp70, Hsp90, Hsp100, and small heat shock proteins with the molecular mass of the subunits varying from 12 to 43 kDa (Hsp20, Hsp22, Hsp25/27, Hsp32, HspB1–HspB10, and others). At the same time, since many chaperones are oligomers, the molecular masses of the functionally active complexes can be significantly higher. For example, Hsp70 class chaperones (molecular mass of 66–78 kDa) function in tandem with Hsp40 co-chaperones, whose mass varies from 10 (DnaJC19) to 254 kDa (DnaJC13). The eukaryotic chaperonin TRiC (Hsp60) is an oligomeric complex (≈1000 kDa) consisting of 16 distinct but homologous subunits. The bacterial chaperonin GroEL (≈800 kDa) consists of 14 identical subunits, but it requires for its functioning co-chaperonin GroES, which is a heptamer (70 kDa).

The folding of newly synthesized polypeptides and refolding of misfolded proteins require chaperones of the HSP60 class (chaperonins) as well as HSP70 and HSP90 chaperones. They are multisubunit complexes, whose functioning includes alternating cycles of binding of non-native proteins and release of folded proteins that are regulated by ATP and various cofactors. The ATPase activity is a distinctive feature of this type of chaperone. At the beginning of the cycle, the chaperones recognize and bind surface-exposed hydrophobic motifs of unfolded proteins to protect them from aggregation. At the end of the cycle, the hydrolysis of ATP results in the release of the protein into the solution, where the folding process must complete spontaneously. The polypeptide chains that have not been folded properly can reassociate with the chaperones. Thus, productive folding requires the participation of the ATP-dependent chaperones, which provide repeated cycles of protein binding and release until it reaches its native state. These chaperones require specific co-chaperones for their functioning that not only regulate the ATPase cycle but also affect substrate specificity [8].

It should be noted that the ATP-dependent chaperones also include chaperonins [14], complexes with the unique two-ring architecture and an internal cavity. The presence of the internal cavity provides an isolated space for the folding of non-native proteins. Due to the difference in their mechanisms, chaperones and chaperonins function sequentially: HSP70 interacts with nascent and newly synthesized polypeptides, and chaperonins are involved in the folding of those proteins that cannot be folded only with the participation of HSP70. At the same time, along with the ATP-dependent chaperones, the aggregation of non-native proteins in the cell, especially under stress conditions, is also prevented by the ATP-independent chaperones, small heat shock proteins, which function as classical holdases [15]. They bind aggregation-prone proteins and transfer them either to ATP-dependent chaperones for refolding or to proteasomes/autophagosomes for proteolytic degradation.

In the following sections, we will consider the interaction of chaperones, including artificial chaperones, with amyloidogenic proteins, the effect of post-translational modifications of both chaperones and their substrates on the pathological transformation of amyloidogenic proteins, as well as the blocking of chaperones by various types of misfolded proteins.

## 3. Effect of Artificial Chaperones on the Pathological Transformation of Amyloidogenic Proteins

Certain functions of protein chaperones can be mimicked using artificial chaperones based on various polymers. The easiest way is to create polymer-based chaperones similar to small heat shock proteins. In this case, it is sufficient to select polymers that bind to unfolded protein molecules in order to prevent their aggregation. Such simple compounds may be even more effective antiaggregant agents than small heat shock proteins due to their higher stability [16]. It is also possible to increase the specificity of the action of artificial chaperones by obtaining conjugates of polymers with antibodies recognizing only non-native forms of proteins [17,18]. In addition, selecting certain combinations of polyelectrolytes makes it possible to imitate the action of more complex chaperones by regulating the dissociation of the polymer–protein complex by changing the ionic strength or pH of the incubation medium [19]. Such rather complex artificial chaperones have not yet been used as reagents preventing the pathological transformation of amyloidogenic proteins. However, the works on the use of various polymers for these purposes are quite numerous (examples are given below).

It has been shown that synthetic and natural polymers can both destroy already formed amyloid structures, such as fibrils, and prevent their formation by binding to monomeric forms of amyloidogenic proteins. Thus, it was found that cationic pyridylphenylene dendrimers [20] and polystyrene sulfonates [21] destroy amyloid aggregates of the recombinant ovine prion. Moreover, certain types of cationic dendrimers bind to monomeric forms of sheep prion [22,23] or beta-amyloid peptide [24], which prevents their further amyloid aggregation. Sulfated aromatic polymers inhibit the formation of alpha-synuclein fibrils in vitro [25], while sulfated polysaccharides inhibit the accumulation of amyloid aggregates of the prion protein in cell culture [26]. However, on the contrary, many polymeric molecules interacting with amyloidogenic proteins stimulate their pathological transformation. For example, natural glycosaminoglycans (heparin, heparan sulfate, and others), as well as proteoglycans containing sulfated polysaccharides, affect the development of amyloid neurodegenerative diseases [27,28,29]. This is possibly due to the fact that heparan sulfate and other glycosaminoglycans stimulate the pathological transformation of amyloidogenic proteins: alpha-synuclein [30], beta-amyloid peptide [31,32], tau protein [33], and apolipoprotein A-I [34].

One more approach to obtain an artificial chaperone is to use a combination of polymers with chaperones. It seems promising to obtain conjugates of small heat shock proteins with polyelectrolytes. Small heat shock proteins increase the specificity of the action of such complexes, and the presence of polyelectrolytes makes it possible to regulate the action of the complexes by changing their solubility by varying the ionic strength or the pH of the medium. In addition, polymeric nanoparticles can be used to deliver chaperones. For example, electro-responsive poly (3,4-ethylenedioxythiophene) nanoparticles were used to deliver small molecule compounds with chaperone-like activity [35]. This method also seems promising for the delivery of protein chaperones, at least small heat shock proteins.

Thus, at present, the use of artificial chaperones as anti-aggregation and anti-amyloid compounds is of empirical nature, and their rational use requires comprehensive studies on the effect of the physicochemical properties of polymers on their interaction with various amyloidogenic proteins, which continue in the laboratory of P. Semenyuk [36].

## 4. Influence of Chaperones on Pathological Transformation of Amyloidogenic Proteins

As noted above, extensive information on the effect of chaperones on amyloid transformations is rather contradictory. In the review, we will mainly consider the effect of chaperones on the interconversions of two amyloidogenic proteins, prion protein and alpha-synuclein, since these proteins were studied in our laboratory.

### 4.1. Alpha-Synuclein

All types of chaperones, from small heat shock proteins to chaperonins, bind to alpha-synuclein and influence its amyloid transformation. Alpha-synuclein does not form a stable tertiary structure and belongs to the so-called intrinsically disordered proteins [37]. Alpha-synuclein is associated with the occurrence and development of a number of synucleinopathies, including Parkinson’s disease [38,39]. A study of the influence of chaperones on the pathological transformation of alpha-synuclein leading to the formation of its oligomeric and fibrillar forms is necessary both for understanding the occurrence of synucleinopathies and searching for approaches to their treatment. Most studies have shown that alpha-synuclein binds to chaperones, and this interaction prevents its amyloidization. Thus, small heat shock proteins (Hsp27 and αB-crystallin) bind to alpha-synuclein and effectively suppress its aggregation and fibrillation [40,41,42,43,44]. More complex ATP-dependent chaperones of the Hsp70 class have an even more pronounced effect on the transformation of alpha-synuclein [45,46,47,48]. The Hsp70 chaperones can prevent the formation of fibrils and oligomers not only in the presence of ATP but also in its absence [46,47]. Some inconsistencies in the results on the effect of nucleotides on the binding of alpha-synuclein are probably associated with the peculiarities of interaction of the chaperones with various intermediate products of alpha-synuclein fibrillation [49]. The chaperone Hsp70 in a combination with J-protein co-chaperones of class B DNAJB1 and HSP110 nucleotide exchange factor efficiently disaggregates already formed amyloid alpha-synuclein fibrils. The disaggregation is ATP-dependent process, and the resulting alpha-synuclein monomers are non-toxic [50].

Other chaperones, including those of the Hsp60 class, can also prevent the formation of fibrils [51,52,53,54]. The eukaryotic chaperonin TRiC binds to alpha-synuclein oligomers, neutralizes their toxicity, and prevents fibrillation [54]. The bacterial chaperonin GroEL reduces the rate of fibril formation [52].

However, some data indicate that chaperones in some cases contribute to the pathological transformation of alpha-synuclein. The chaperonin Hsp90 was shown to bind to the monomeric alpha-synuclein in vitro, and its further transformations depended on the functional state of the chaperonin. In the absence of ATP, the chaperonin stimulated the accumulation of non-amyloid alpha-synuclein oligomers, while the formation of alpha-synuclein fibrils was an ATP-dependent process [55]. As noted above, the combination of three chaperones of multicellular organisms, HSP70, DNAJB1, and HSP110, resulted in disaggregation of alpha-synuclein fibrils in vitro, yielding non-toxic monomers. At the same time, the functioning of these chaperones in vivo probably stimulated the appearance of toxic forms of alpha-synuclein. In experiments on *Caenorhabditis elegans*, the depletion of HSP-110 decreased the HSP70 disaggregation activity, which resulted in reducing alpha-synuclein foci, cell-to-cell transmission, and toxicity [56]. Therefore, although the HSP70 chaperone and its partners are required to maintain cellular proteostasis, they can be involved in the formation of toxic forms of alpha-synuclein. Thus, although in most cases, chaperones prevent the pathological transformation of alpha-synuclein and even destroy already formed amyloid structures, in some cases, they can promote the transmission of toxic oligomeric species of alpha-synuclein formed during the disaggregation of fibrils.

### 4.2. Prion Protein

Different types of chaperones are involved both in the formation of the native prion protein (PrPc), which performs its functions in the nervous system, and in the pathological transformation of this protein. The chaperones of bacterial cells of the gastrointestinal tract, which are involved in the transformation of the infectious prion protein and its transport to the central nervous system, may be of particular importance for the development of infectious forms of prion diseases [57]. The mammalian prion protein consists of two fundamentally different structural domains: an unstructured N-terminal domain and a structured C-terminal domain containing three alpha-helices and two beta-strands [58]. The prion protein is synthesized in the rough endoplasmic reticulum (ER) and travels through the Golgi apparatus to eventually reach the cell surface. During biosynthesis in the ER, the prion protein undergoes several post-translational modifications: cleavage of the N-terminal signal peptide, the addition of oligosaccharide chains at two sites at the N-terminus, the formation of a single disulfide bond, and the attachment of a glycophosphatidylinositol anchor (GPI) to the C-terminus of the protein [59,60,61]. After maturation, the protein is attached to the cell surface using GPI [60].

It is known that in mammals, the prion protein is expressed mainly in neurons and is localized on the membranes of cells that form the diffuse neuroendocrine system and the cells of the lymphoreticular system [62]. The prion protein also can be detected in the cytosol of neurons [63,64]. In more detail, the processes of protein translation, their proper folding, maturation, and transport take place in the endoplasmic reticulum (ER) with the participation of chaperones [65,66]. A disruption of protein folding in the cell results in the accumulation of misfolded proteins in the endoplasmic reticulum leading to endoplasmic reticulum stress (ER stress) [67]. The overloading of ER with unfolded proteins activates an unfolded protein response (UPR), which is aimed to restore normal cell function by stopping protein translation, cleaving misfolded protein molecules, and activating the synthesis of molecular chaperones [68]. If the cell fails to restore its functionality, the UPR directs it along the path of apoptosis [69]. It was shown that the ER stress is often observed when working with models of prion diseases and makes a significant contribution to the development of these pathologies [70,71,72]. Moreover, the ER stress has been observed in patients with the sporadic and variant forms of Creutzfeldt–Jakob disease [73]. One of the significant participants in the ER stress is the eukaryotic chaperone BiP/Grp78 [74], which is a member of the Hsp70 family [75]. It is one of the most abundant proteins in the ER, making it the main driver of protein folding in the cell [76]. Grp78 interacts with mutant PrP and mediates its degradation through the proteasome, which may indicate that Grp78 accompanies prion protein folding during its de novo synthesis [77,78]. The level of Grp78 increased in the neuroblastoma cells infected with the scrapie isoform of the prion protein PrPSc [73,79] as well as in mice infected with the prion protein [80]. More importantly, brain samples from patients with sporadic Creutzfeldt–Jakob disease were found to have elevated levels of this particular chaperone [73]. It is likely that the chaperone Grp78 is involved in the folding of the native prion protein. The accumulation of mutant or infectious forms of the prion protein disrupts the functioning of the Grp78 chaperone and causes ER stress, which is accompanied by the stimulation of Grp78 synthesis. The chaperones Hsp72 and Hsp73 are also associated with prion diseases. It was found that the brain of mice with scrapie contains an abnormally large number of lysosomes enriched in PrP and Hsp73 [81]. The content of Hsp72 increases in neurodegenerative diseases [82] as well as in the modeling of prion diseases both in cells [83] and in animals [84].

Recent studies including the works of our group have visualized the interaction of monomers of a prion protein cellular isoform with the bacterial chaperonin GroEL (Figure 1) [85]. This interaction with GroEL in the absence of GroES leads to the amyloid transformation of PrPc, resulting in the formation of large amyloid aggregates [57]. At the same time, the complex GroEL-GroES is also able to bind to monomers, toxic oligomers, protofibrils, and prion protein fibrils. The chaperonin GroEL-GroES in the presence of ATP promotes amyloid transformation of the monomeric and oligomeric forms of PrP but also results in partial disassembly of PrP amyloid fibrils [86]. However, the eukaryotic chaperonin TRiC, the apical domains of which perform a function of GroES, is not capable of disassembling oligomeric prion forms. The interaction of this chaperonin with various forms of the prion protein leads to their enhanced amyloid aggregation [57,87]. Thus, the effect of different chaperonins on the amyloid transformation of prions is rather specific, and it is difficult to predict due to the influence of numerous factors.

It should be noted that the role of chaperones in the transformation of yeast prion proteins has been studied in detail [88]. Studies have shown that the yeast chaperone Hsp104 is involved in the cleavage of amyloid fibrils formed from the yeast prion protein Sup35 [89]. Although such yeast models are distantly related to the mechanisms of the occurrence of prion diseases in animals, the ability to carry out genetic manipulations with yeast cells makes it possible to quickly and easily identify the general patterns of the effect of chaperones on amyloid transformation of proteins [90,91,92]. In addition, the expression of mammalian amyloid proteins in yeast cells allows the approximation of these models to those based on mammalian cells [93].

Summarizing, we can say that chaperones play an important role both in the implementation of the natural functions of the amyloidogenic prion protein and in its pathological transformation. Chaperones are involved in the folding of the prion protein and in its transport to the membranes of the nerve cells. At the same time, mutant or infectious forms of prions can block chaperones, causing ER stress. Chaperones are involved in the destruction of already formed oligomeric and fibrillar forms of the prion protein [86]. In addition, bacterial chaperones may be involved in the transport of infectious forms of prions from the gut to the central nervous system, which will be discussed in more detail in the last section of the review.

Information about the influence of different groups of chaperones on the pathological transformation of two amyloidogenic proteins—prion protein and alpha-synuclein—is summarized in Table 1. As follows from the table, chaperones prevent the aggregation and fibrillation of both prion protein and alpha-synuclein. Some chaperones can even destroy already formed amyloid fibrils. At the same time, such destruction of fibrils cannot always be referred to as the favorable effect of chaperones, since more and more data point to the toxicity of precisely small oligomeric amyloid structures. In addition, in some works, chaperones have been shown to stimulate the pathological transformation of amyloidogenic proteins. We believe that this change in the function of chaperones can be explained by the influence of the following factors: blocking the chaperones by misfolded proteins (Section 6) or the loss of the ATPase or protein-binding activity, which could be due to their post-translational modifications (Section 7).

## 5. Mechanisms of the Formation of Misfolded Proteins

To understand the features of the functioning of chaperones, the key issue is the mechanisms of recognition of the native and misfolded protein structures. The simplest model consists in the presence of only two structures: a fully or partially unfolded polypeptide chain that binds to the chaperone and a native protein molecule that does not interact with the chaperone. In this case, it is assumed that the unfolded protein molecule must fold into a strictly defined native conformation. However, in reality, the process of protein folding is much more complicated. Discussions about the relationship between the primary structure of a protein and the emergence of a native protein conformation have been going on for many decades. The initial ideas that the polypeptide chain should take a strictly defined conformation, which corresponds to the global minimum of the Gibbs free energy, began to be criticized [94,95,96,97,98].

To date, many papers have been published in which the value of the Gibbs free energy of the protein folding process is determined from experiments on denaturation and subsequent renaturation [99,100,101,102,103]. The main controversial point of this method is the assumption that proteins are completely denatured under these experimental conditions, and such denaturation is completely reversible. However, a detailed analysis of the protocols of such experiments shows that most of the methods for assessing the unfolding of the protein structure are indirect and replete with numerous assumptions [99,100,101,102,103,104,105,106]. Thus, proteins that are supposed as fully unfolded are actually only partially unfolded, and the degree and nature of unfolding of the same protein can differ significantly depending on the denaturation protocol used [104,107]. Therefore, almost all of the experimental data cited in support of the spontaneous protein refolding hypothesis have been obtained on a limited set of compact, stable, single domain monomeric proteins whose folding ΔG ranges between −3.5 and 7 kcal/mol [97,108,109]. However, most proteins are not capable of spontaneous folding and, once fully unfolded, are not capable of subsequent refolding [110,111,112].

The problem of folding often arises during the production of a recombinant protein in bacterial cells, when overexpressed recombinant proteins form insoluble inclusion bodies [20,113,114,115]. Such proteins can be successfully converted into a soluble form with the restoration of their activity after purification from other cellular and protein fractions. As a rule, proteins from inclusion bodies are not initially unfolded or disordered. On the contrary, they have well-defined ordered secondary and tertiary structures, which are often enriched in beta-sheets [116,117]. It should be noted that even in the case of successful refolding, it hardly resembles the native folding processes that occur in living cells. The purification and refolding of almost every such protein requires the development of a separate protocol, which often uses conditions that are far from physiological, and the refolding time is usually much longer than the biologically significant time values.

More than that, the cell contains large concentrations of macromolecules (up to 400 g/L), among which are proteins, nucleic acids, lipids, glycans, and solvated ions [118], which suggests that approximately 40% of the cell volume can be occupied by macromolecules and become physically inaccessible to other molecules. Accordingly, it can be assumed that crowding conditions affect both the kinetics and thermodynamics of interactions between macromolecules, including protein folding and aggregation [119,120,121]. Since crowding has a strong effect on protein–protein interactions, its influence on the conformation and self-association of chaperones, interaction of chaperones with target proteins, and the aggregation of the target proteins should also be taken into account [122].

Thus, the article by Koonin [123] and the previous works of his co-authors [108,124,125,126] consider an alternative hypothesis of protein folding, the kinetic one, according to which the native conformation of most proteins is not in the global but rather at a local minimum in the fluctuating free energy landscape. Moreover, the free energy values are probably positive for most proteins, which implies the energy costs to adopt a native conformation, which is only possible as a result of the interaction of these proteins with the molecular machines of the cell, such as translation systems or chaperones.

Protein engineering experiments show that many proteins are quite easily destabilized by introducing small substitutions into their sequence. However, despite many years of research, predicting the effect of mutations on protein stability remains a difficult task, and much less attention is paid to various post-translational modifications of proteins, which can be subjected not only to the formed protein globule but also to the original polypeptide chain. For example, a mutation on the surface of a protein globule that increases its solubility is often destabilizing [127]. Similarly, mutations that increase the activity of an enzyme often destabilize the protein, and vice versa, stabilizing mutations often decrease its activity [128].

Thus, it becomes obvious that any post-translational modification can affect folding in such a way that the protein, in principle, cannot adopt its native conformation. This remark also applies to the folding of mutant recombinant proteins. In the latter case, researchers often try not to notice that the introduction of a mutation can not only affect the functioning of the active center of the enzyme but simply lead to its inaccurate or incomplete folding. In addition, a protein can lose its solubility and will be produced immediately into inclusion bodies, as we discussed above. It should also be noted that various post-translational modifications can shift the balance between different protein conformations.

A separate issue is the general instability of the cellular proteome that is the widespread spontaneous loss of the native structure and activity of proteins that were initially correctly folded. This loss of native conformation and functional activity occurs both under conditions of isolated proteins in vitro and in the cell system in vivo. It is known that all living cells maintain complex mechanisms of proteostasis to repair or destroy damaged proteins that have irreversibly lost their native form [129,130,131]. Disturbances in the utilization of misfolded and/or inactive proteins lead to various types of disorders (often fatal) of cellular homeostasis, including the deposition of protein aggregates and the subsequent development of various diseases [129,130].

Thus, there is a possibility of the emergence of several conformational states of a protein or the appearance of non-native conformations due to mutations or post-translational modifications as well as due to changes in the environment in which the protein is folded. The chaperones make it possible to change an incorrect protein conformation due to alternating cycles of binding/release of proteins or to facilitate folding by changing the environment in which folding occurs.

This is precisely the function of complex multisubunit chaperonins; in the internal cavity, conditions are created that exclude the aggregation of polypeptide chains or their interaction with other macromolecules. However, polypeptide chains with mutated or modified amino acid residues may not always acquire a native conformation even under such “ideal” conditions. The features of the interaction of this type of misfolded protein with chaperones will be considered in the next section.

## 6. Blocking of Chaperones by Misfolded Proteins and Amyloidogenic Proteins

Chaperones contribute to the correct folding of polypeptide chains, which leads to the appearance of functionally active proteins with a native conformation. Probably, as it was mentioned in the previous section, several native conformations are possible, including those with different functions, but when considering the blocking of chaperones, we will not touch on this aspect. We will assume that there are incorrectly folded or unfolded polypeptide chains that bind to chaperones and proteins in native conformation that are not recognized by chaperones (Figure 2). A protein molecule with a native conformation appears after one or more cycles of binding of an unfolded polypeptide chain to a chaperone and the subsequent dissociation. However, if the polypeptide chain cannot fold into its native conformation in principle, it will again bind to the chaperone. In the case of complex chaperonins, where the polypeptide chain folding occurs in the internal cavity, such misfolded proteins will not release into the solution. Such proteins will re-bind to the chaperone sites that recognize structural elements specific to unfolded proteins. Consequently, the misfolded proteins discussed above that arise for various reasons will block the chaperone system (Figure 2). In the case of complex chaperonins, such as GroEL and TRiC, the blocking of even one ring should disrupt the alternate operation of the two rings and completely inhibit the function of the chaperonins. Blocking of the chaperonin GroEL has been studied in detail using modified or mutant forms of glyceraldehyde-3-phosphate dehydrogenase (GAPDH) [132,133]. It was shown that the O-R-type mutant dimers of GAPDH strongly bind to GroEL in a 1:1 molar ratio and completely block its ability to reactivate proteins. The mutant O-R-type dimers of GAPDH from *Bacillus stearothermophilus* are formed by introducing substitutions Y283V, D282G, or Y283V/W84F. At the same time, the O-P-type dimers of the mutant GAPDH formed upon substitutions Y46G/S48G and Y46G/R52G do not bind to the chaperonin and do not affect its functional activity [132]. Modified forms of GAPDH from rabbit muscles have a similar effect on the chaperonin GroEL. GAPDH modified by dithiobisnitrobenzoate at the sulfhydryl groups binds to GroEL and blocks its activity against unmodified GAPDH polypeptide chains [132]. In addition, the oxidation of unfolded GAPDH polypeptide chains with hydrogen peroxide yields the protein molecules that are unable to fold into their native conformation. Such oxidized polypeptide chains of GAPDH strongly bind to one of the GroEL rings and completely block its activity against unmodified forms of GAPDH denatured in guanidine hydrochloride [133]. Interestingly, such blocked chaperone retains the ability to reactivate denatured lactate dehydrogenase, which is probably due to its binding to the apical domain of the opposite GroEL ring with their subsequent release into the solution without encapsulation (trans-mechanism) [133]. Thus, we can conclude that the blocking of chaperones can occur when they bind to mutant or modified forms of proteins that cannot adopt a completely native conformation and are recognized by chaperones as proteins containing non-native motifs (Figure 2).

The blocking of chaperones may occur when they interact not only with misfolded proteins but also with amyloidogenic proteins (Figure 2). Amyloidogenic proteins are intrinsically disordered proteins, and therefore, they contain unstructured regions that are recognized by chaperones as non-native structures. Most studies focus on the role of chaperones in the pathological transformation of amyloidogenic proteins. However, amyloidogenic proteins can also affect chaperones by blocking their active centers (Figure 2). Thus, it was shown that prion protein monomers bind in the internal cavity of the GroEL chaperonin. Such binding decreases the flexibility of the chaperonin ring in which the prion protein is bound [85]. The interaction of PrPc with GroEL disturbs the functioning of the GroEL–GroES complex: the chaperonin loses its ability to reactivate denatured GAPDH [86]. The same effect was observed in experiments with the eukaryotic chaperonin TRiC. The prion protein completely blocked the ability of TRiC to reactivate the denatured recombinant human sperm-specific glyceraldehyde-3-phosphate dehydrogenase (GAPDS). Glycation of the PrPc monomer resulted in the partial unfolding of its structured part, which decreased the efficiency of the chaperonin blocking by PrP [87].

Blocking the chaperone system, which performs such important functions for the cell, by amyloid proteins should lead to a change in the functional proteome of the cell. It is well known that neurodegenerative amyloid diseases are characterized not only by the appearance of oligomeric and fibrillar amyloid structures but also by various disturbances in cell functioning. For example, Parkinson’s disease is accompanied by disorders in energy metabolism, including glycolysis. One of the possible mechanisms for reducing the rate of glycolysis may be the lack of glycolytic enzymes, which constitute the majority of cytoplasmic proteins. Amyloid forms of alpha-synuclein can not only directly inhibit the activity of glycolytic enzymes, as was recently shown [134], but also block the chaperones that are necessary for efficient folding of the enzymes. In addition, the blocking of chaperones by amyloid proteins can also impair other cell functions due to the inability to produce sufficient amounts of proteins in their native conformations. All little-studied aspects of the effect of blocking chaperones by amyloid and misfolded proteins on cell functioning require close study.

Post-translational modifications of both amyloidogenic and globular proteins can play a special role in the blocking of chaperones. Post-translational modifications of amyloidogenic proteins, as a rule, weaken their binding to chaperones [87,135,136], which prevents their blocking effect on the chaperone system. One of the main mechanisms for the accumulation of modified misfolded proteins in the brain in pathological disorders is exposure to reactive oxygen species (ROS) and reactive nitrogen species (RNS) [137]. It should be noted that RNS modify proteins not only by S-nitrosylation of their sulfhydryl groups. For example, alpha-synuclein, which does not contain cysteine residues, can be subjected to nitration at tyrosine residues, which enhances its aggregation [138].

At the same time, post-translational modifications of globular proteins, which are not characteristic of their native conformations, on the contrary, stimulate the blocking of chaperones due to the impossibility of folding the polypeptide chain into the native conformation [132,133].

We believe that the study of the mechanisms of chaperone blocking will make it possible to find rational ways to increase the efficiency of systems involved in protein folding. So, instead of complex methods of increasing the concentration of chaperones by stimulation of their expression in cells or adding the chaperones from the outside, it could be useful to prevent post-translational modifications of proteins leading to the accumulation of misfolded proteins. For example, the use of antioxidants can prevent the oxidative modification of proteins, thereby reducing the appearance of chaperone-blocking protein forms.

## 7. Post-Translational Modifications and Functioning of Chaperones

In the previous sections, it was noted that the consequences of the interaction of amyloidogenic proteins with chaperones depend on the functional state of the latter. The functioning of chaperones can be impaired not only by their blocking by misfolded proteins described in the previous section but also by post-translational modifications. As a rule, modifications of chaperones lead to a decrease in their anti-aggregation activity. For example, the modification of small heat shock proteins, causing the formation of cross-linked oligomeric forms, reduces their anti-aggregation activity [139]. Complex ATP-dependent chaperones are particularly sensitive to such modifications [137]. The chaperonin Hsp90 can be nitrated at tyrosine residues, which induces nerve cell death [140]. An increase in the level of S-nitrosylated Hsp90 observed in cardiac hypertrophy in mice is mediated by a decreased expression of S-nitrosoglutathione reductase [141]. It was shown that the S-nitrosylation of Hsp90 inhibits its ATPase activity [142]. S-nitrosylation of the bacterial co-chaperone DnaJ leads to its inability to interact with the Hsp70 chaperone DnaK in mediating the correct folding of denatured rhodanese [143]. Modifications of chaperones can also decrease their affinity for substrate proteins. For example, trimethylation of the Hsp70 chaperone decreases its affinity for both the monomeric and fibrillar forms of alpha-synuclein [144].

The most serious, in our opinion, are the consequences of the modifications, in which chaperones retain the ability to bind amyloid proteins but lose the ATPase activity. In such a situation, the dissociation of the chaperone–amyloid protein complex during the ATP-dependent cycle is impossible. As a result, the chaperone without ATPase activity can stimulate the formation of fibrils that are not formed by native chaperones [57]. Therefore, such a modification completely changes the function of the chaperone, converting it from an anti-amyloid protein into a protein that provokes amyloidogenesis.

## 8. Participation of Bacterial Chaperones of Gut Microbiota in the Pathological Transformation of Amyloidogenic Proteins

Over the past decade, a lot of information has appeared on the relationship between the processes occurring in the gastrointestinal tract and the emergence and development of neurodegenerative diseases, primarily Parkinson’s and Alzheimer’s diseases [145,146,147,148,149]. The determining role in the influence of the gastrointestinal tract on the nervous system is played by the gut microbiota, the composition of which can both provoke pathological processes and prevent them. The participation of the gut microbiota in the development of Parkinson’s disease has been studied most fully [149,150,151,152,153,154]. First of all, several studies have shown that alpha-synuclein and its fibrillar forms can be transported from the gut to the brain and participate in the formation of Lewy bodies [151,152]. In this case, both alpha-synuclein of endocrine cells of the intestinal wall [151] and fibrillar forms of alpha-sinuclein introduced into the intestinal lumen can be transported [152]. Moreover, it has been shown that intestinal nerve cell alpha-synuclein is altered by gut-derived metabolites. As a result, alpha-synuclein can not only acquire a pathological conformation itself but also penetrate into the central nervous system tissues and change the conformation of alpha-synuclein in them. Thus, it was shown that the altered forms of alpha-synuclein have a prion-like effect, which may be one of the mechanisms for the development of this pathology [153,154]. It is generally accepted that it is the various metabolites formed in the gut that can penetrate the intestinal wall and affect the processes occurring in the central nervous system. However, we believe that another mechanism, proposed by us in 2011, is also possible, which primarily can be characteristic of infectious prion diseases [57]. Infectious prions enter the body through the gut and, of course, can meet with bacteria in it, as well as with components released during the lysis of bacterial cells. These components can include chaperones of both bacteria themselves and bacteriophages. Chaperones such as GroE are known to interact with prion proteins and influence their pathological transformation (see Section 4.2). We have shown that the GroE chaperonin from *E. coli* cell extracts stimulates the amyloid transformation of the recombinant ovine protein and causes the formation of spherical nanostructures. The efficiency of the amyloid transformation correlates with the content of GroE in cells, since the maximal accumulation of amyloid forms of the prion protein is typical for extracts of superproducers of this chaperone [57]. We believe that the chaperonins of bacteria, bacteriophages, and other inhabitants of the gut may be a key element that determines the efficiency of infecting an organism with the prion protein. It cannot be ruled out that other amyloidogenic proteins that enter the gut with food, for example, alpha-synuclein, after interaction with chaperones, can penetrate into the central nervous system and provoke pathological transformation of proteins associated with neurodegenerative amyloidosis. This almost unexplored aspect of the role of the gut microbiota requires careful study, as it allows us to establish the molecular mechanisms of such a currently popular phenomenon.

## Figures and Tables

**Figure 1 ijms-23-02747-f001:**
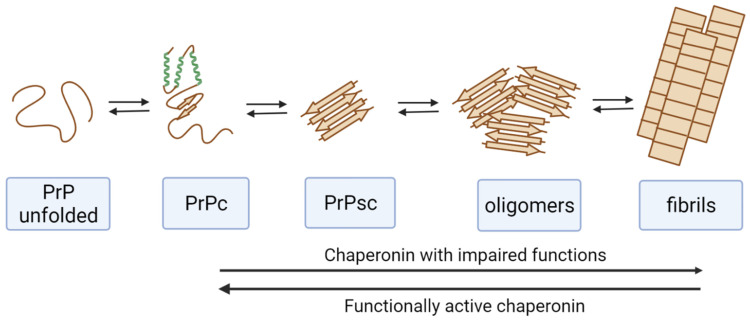
Effect of chaperonins on the pathological transformation of the prion protein (PrP). Functionally active chaperonins perform the disassembly of amyloid forms of PrP. Chaperonins with impaired functions (modified, blocked with misfolded proteins, etc.) stimulate PrP amyloid transformation.

**Figure 2 ijms-23-02747-f002:**
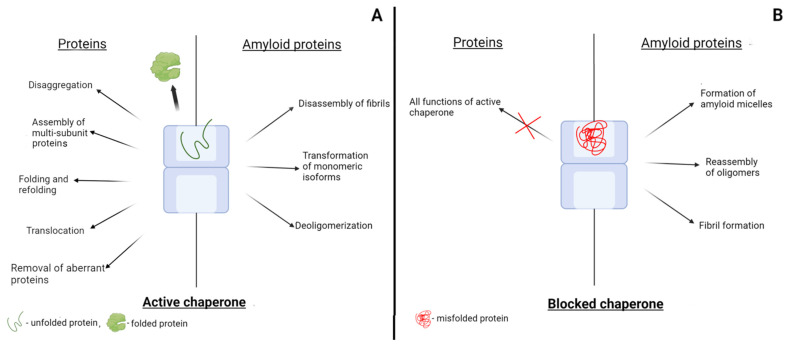
Changes in the function of chaperones driven by the blocking with misfolded forms of proteins. (**A**) The effect of chaperones on the conversion of various proteins and on the transformation of amyloid proteins. (**B**) Induction of pathological transformation of amyloidogenic proteins by blocked chaperones.

**Table 1 ijms-23-02747-t001:** Effect of different chaperones on the transformation of alpha-Synuclein (A-Syn), ovine prion protein (PrP), and yeast prion protein (Sup35p).

Types of Chaperones	Molecular Mass of One Subunit, kDa	Number of Subunits	ATPaseActivity	Inhibition of Aggregation or Fibrillation	Stimulation of Fibrillation	Fibril Disaggregation
Small heat shock proteins	12–43	9–50	-	A-Syn (40–44)		
Hsp70	66–78	1–2	+	A-Syn (45–48)PrP (77)		A-Syn(50, 56)
Hsp90	81–99	2	+		A-Syn (55)	
Hsp100–110	100–199	6	+			A-Syn (50, 56)Sup35p (89)
GroEL (Hsp60)	55–64	14	+	A-Syn (51–53)	PrP (57)	
GroEL + GroES			+		PrP (86)	PrP (86)
TRiC (Hsp60)	55–64	16	+	A-Syn (54)	PrP (87)

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
