# Peer review of "Regulation by Different Types of Chaperones of Amyloid Transformation of Proteins Involved in the Development of Neurodegenerative Diseases"

_ijms, 2022, doi:10.3390/ijms23052747_

Round 1

Reviewer 1 Report

This is an unbalanced review.

The small type writing should be before in section 2. From general description what are chaperones and their function to the specific types.

 The last part of this section should be kept to introduce the next section. 

 A table could elucidate a structure/function relationship.  

To add clarity: a figure showing an HSP: unfolded, folded, misfolded

Provide structure of simple vs complex chaperone

Better explain artificial beneficial vs pathogenic chaperone - what makes the difference for prion and beta amyloid 

The other sections better structured but the discussion in each section should be better organized - many papers quoted but putting a concluding statement what was learned and what is next step in enquiry would be beneficial.

The misfolded protein section is too long and fragmented. (section 5)

  1. HSPs that are beneficial/pathologic
  2.  

Author Response

We are grateful to the Reviewers for their comments and suggestions. We have made the necessary amendments to the text, which we hope will improve our article. Below are the responses to the Reviewers. In the text of the article, the changes made are highlighted in yellow.

Reviewer 1

1) This is an unbalanced review.

We have adjusted the text of the review in accordance with the suggestions below.

2) The small type writing should be before in section 2. From general description what are chaperones and their function to the specific types. The last part of this section should be kept to introduce the next section. 

As suggested by the Reviewer, the second section begins with a general description of chaperones, followed by a brief classification of chaperones, and a concluding phrase that precedes the following sections.

 3) A table could elucidate a structure/function relationship.  

A table has been added that includes a list of chaperones considered in the review, their main structural features, as well as the effect of the chaperones on the aggregation and transformation of amyloidogenic proteins.

4) To add clarity: a figure showing an HSP: unfolded, folded, misfolded

In Figure 2, an image of a folded protein resulting from the action of a chaperone is added, as well as images of unfolded, folded, misfolded proteins with corresponding captions.

5) Provide structure of simple vs complex chaperone

The review mentions many different chaperones that differ in their structure, which is difficult to depict in one figure. In addition, many reviews and books summarize these data. We decided to present a table 1 that includes the main structural features of chaperones (molecular weight of subunits and the number of subunits in a functionally active chaperone), as well as information on the effect of chaperones on the transformation of amyloidogenic proteins.

6) Better explain artificial beneficial vs pathogenic chaperone - what makes the difference for prion and beta amyloid

We have edited this section. We clarified that prion protein also belongs to amyloidogenic ones and tried to more accurately describe the effect of chaperones on various amyloid proteins (prion protein and alpha-synuclein). We hope that the added table will also make this question clearer. Unfortunately, chaperones have a dual effect on the pathological transformation of amyloidogenic proteins, which we are trying to emphasize once again in this review. Most likely, there are no fundamental differences in the action of the chaperone on the prion protein and other amyloid proteins. In this case, as described in the following sections, the effect of the action of chaperones will depend on their state (functionally active, blocked chaperone or chaperone without ATP and other components). In addition, it is very difficult to unambiguously answer this question, since there are still disputes about the toxicity of various forms of amyloidogenic proteins. Therefore, the ability of chaperones to destroy amyloid fibrils can be considered a pathogenic effect if the high toxicity of oligomeric forms is finally established.

7) The other sections better structured but the discussion in each section should be better organized - many papers quoted but putting a concluding statement what was learned and what is next step in enquiry would be beneficial.

The concluding statements have been added to sections where they were absent.

8) The misfolded protein section is too long and fragmented. (section 5).

Section 5 has been shortened and edited.

Sincerely yours,

Vladimir Muronetz.

Reviewer 2 Report

The authors of this rewiev report a broad overview of the role played by a
class of proteins, the chaperones, in the development of neurodegenerative
diseases. In particular, they articulate the rewiev by first illustrating the
characteristics and roles played by the different classes of chaperones in the
possible development of neurodegenerations. They consider the role played by
chaperones on the transformation of amyloid proteins considering the role of
synuclein and prion protein; they deal with the mechanism underlying the
misfolding of proteins associated with amyloid degeneration, referring to a
good part of the work done in their studies. One piece of information that
I find very interesting is the possibility of creating artificial chaperone
proteins, although the data on this are not very satisfactory.
In this regard, I ask the authors if they have thought about the creation
of nanoparticulate structures that can be effective for binding selected
chaperone proteins and targeting them specifically.
Overall, the rewiev is presented in an understandable form and with a
good degree of references that make it acceptable for publication.

Author Response

We are grateful to the Reviewers for their comments and suggestions. We have made the necessary amendments to the text, which we hope will improve our article. Below are the responses to the Reviewers. In the text of the article, the changes made are highlighted in yellow.

Reviewer 2.

1) The authors of this rewiev report a broad overview of the role played by a class of proteins, the chaperones, in the development of neurodegenerative diseases. In particular, they articulate the rewiev by first illustrating the characteristics and roles played by the different classes of chaperones in the  possible development of neurodegenerations. They consider the role played by chaperones on the transformation of amyloid proteins considering the role of synuclein and prion protein; they deal with the mechanism underlying the misfolding of proteins associated with amyloid degeneration, referring to a good part of the work done in their studies. One piece of information that I find very interesting is the possibility of creating artificial chaperone proteins, although the data on this are not very satisfactory. In this regard, I ask the authors if they have thought about the creation of nanoparticulate structures that can be effective for binding selected chaperone proteins and targeting them specifically. Overall, the rewiev is presented in an understandable form and with a good degree of references that make it acceptable for publication

We are grateful to the Reviewer for an interesting proposal concerning the preparation of nanoparticles that could be used for targeted delivery of chaperones. We have not been engaged in obtaining such particles, although there are some works on the use of nanoparticles for the delivery of low-molecular-weight compounds with chaperone-like activity. A reference to such a work has been added to the article. As mentioned in the review, we used antibody-conjugated polymers to generate artificial chaperones. The approach seems to be promising for obtaining conjugates of polymers with small heat shock proteins. In this case, the polymeric component can be used both as a carrier for the chaperone and for regulating its action. In the future, we will try to implement the reviewer's proposal in our work.

Sincerely yours,

Vladimir Muronetz.

Round 2

Reviewer 1 Report

At this point this review is acceptable for publication.

The subject matter is complex but it should encourage further studies since Neuroinflammation/damage still in many cases remain a black box where effective therapies are lacking.